# Drug-Related Deaths in a Tertiary Hospital: Characteristics of Spontaneously Reported Cases and Comparison to Cases Detected from a Retrospective Study

**DOI:** 10.3390/jcm10184053

**Published:** 2021-09-08

**Authors:** Ana Lucía Arellano, Pau Alcubilla, Magí Farré, Eva Montané

**Affiliations:** 1Department of Pharmacology, Therapeutics and Toxicology, Universitat Autònoma de Barcelona, 08001 Bellaterra, Spain; alarellano@clinic.cat (A.L.A.); mfarre.germanstrias@gencat.cat (M.F.); 2Department of Clinical Pharmacology, Germans Trias i Pujol University Hospital, 08916 Badalona, Spain; 3Department of Clinical Pharmacology, Hospital Clinic of Barcelona, 08036 Barcelona, Spain; alcubilla@clinic.cat

**Keywords:** drug-related deaths, spontaneous reporting, active surveillance, pharmacovigilance, adverse drug reaction, hospital mortality

## Abstract

Drug-related deaths (DRDs) are a common cause of hospital death. Pharmacovigilance, either as spontaneous reporting or active surveillance, plays a key role in the detection and reporting of suspected adverse drug reactions (ADRs). We conducted a retrospective analysis of all DRDs spontaneously reported to a pharmacovigilance program of a tertiary hospital, by health care professionals. We compared these results to those of a previous retrospective study conducted in the same hospital from the hospital’s mortality registry. From 1460 spontaneous reported ADRs in a 10-year period, 73 (5%) were DRDs. The median age of DRD was 75 years (range 1 month–94) and 60.3% were men. The most frequent DRDs were hemorrhages (41.1%), followed by infections (17.8%). The most frequently involved drugs were anticoagulants and/or antithrombotic (30%), and antineoplastics (26.3%). When comparing both studies, spontaneous reporting detected more type B reactions (*p* < 0.001) and hospital-acquired DRD (*p* < 0.001); the number of concomitant drugs was higher (*p* = 0.0035); and the kind of ADR were different. The combination of several methods is mandatory to detect, assess, understand, and design strategies to prevent ADRs in a hospital setting, to ensure patient safety.

## 1. Introduction

The definition of an adverse drug reaction (ADR) has evolved during time, it can be defined as ‘any appreciably harmful or unpleasant reaction, resulting from an intervention related to the use of a medicinal product, which predicts hazard from future administration and warrants prevention or specific treatment, or alteration of the dosage regimen, or withdrawal of the product’ [1,2,3,4]. ADRs have high clinical, social and economic burden as a result of a life-threatening risk, stopping an effective treatment, additional medical interventions, use of health services, and long hospitalizations [5,6,7]. Moreover, ADRs are also an important cause of mortality. An ADR could occur in 10% of outpatients, can cause 5–10% of hospital admissions, and could occur in 10–20% of hospitalized patients, prolonging hospital stay [6,8,9]. In the hospital setting, ADRs are the most common safety-concerning events. Drug-related deaths (DRDs) are the fifth most common cause of hospital death [10]. In Europe, about 197,000 deaths appear annually due to an ADR, according to extrapolated data from a meta-analysis [11]. In a recent meta-analysis, in European hospitals, DRD occurrence rates were 7.3% among deceased inpatients and 0.13% among hospitalized patients [12].

Although drugs are recognized as a major cause of mortality, data on in-hospital DRD characteristics is limited. Two meta-analyses of DRDs that led to hospitalization and occurred during admission reported that the most common DRDs were hemorrhages, renal failure, and cytopenia. Anticoagulants, anti-thrombotics, nonsteroidal anti-inflammatory drugs (NSAIDs), renin-angiotensin system inhibitors, and digoxin were the most involved drugs [13,14] Another recent meta-analysis reported that the most common ADR related to death were hemorrhages due to antithrombotic drugs and infections in drug-immunosuppressed patients [12].

Pharmacovigilance, the science and activities relating to the detection, assessment, monitoring, and prevention of ADRs, plays a key and major role in ensuring patient safety [5,6,15,16]. Therefore, the detection and reporting of any suspected ADRs in clinical practice is important to monitor and possibly prevent DRD.

Principally, a pharmacovigilance program may include, but is not limited to, ADR “spontaneous reporting” (SR) from health professionals in clinical practice. This is the most traditional and simple ADR reporting method, but it has been proven to be insufficient, mostly due to under-reporting [17,18]. Other pharmacovigilance methods, known as “active surveillance” (AS), use, for example, chart review, clinical records, ward rounds, telephone structured interviews, or computer monitoring. The AS gives more-detailed information and data about an ADR, but the burden is substantial, requiring trained reviewers [17].

In general, the corner stone of pharmacovigilance programs is the spontaneous reporting method. In order to see if a combination of pharmacovigilance methods could add new information on DRD characteristics in hospitalized patients, this study was designed.

The main objectives of this study were to describe the characteristics of DRD from a hospital pharmacovigilance program that included ADR spontaneously reported by health professionals in our center and, additionally, to compare them to those of a retrospective study, considered as an active surveillance method, carried out by clinical pharmacologists from a systematic selection of DRD cases.

## 2. Materials and Methods

### 2.1. Study Setting and Design

The Germans Trias i Pujol University Hospital in the Barcelonès Nord i Maresme area of Barcelona is a 516-bed tertiary care hospital.

The pharmacovigilance program of the hospital includes the traditional spontaneous reporting method in which ADRs are notified either by nurses, doctors, medical students, pharmacists, or clinical pharmacologists. All the reported ADRs are accurately evaluated by the Pharmacovigilance Committee composed by clinical pharmacologists and specialized nurses. When cases were considered possible, probable, or definite of ADRs, they were included in the pharmacovigilance registry. Moreover, the pharmacovigilance program also includes specific retrospective or prospective studies using an active surveillance method, which are carried out by clinical pharmacologists and specialized nurses as part of the Clinical Pharmacology Department activities. All the registered ADRs are reported to the Spanish Pharmacovigilance System.

We retrospectively analyzed selected spontaneous reported DRD cases from the hospital´s pharmacovigilance program registry. The selection period was from January 2009 to December 2018. DRD cases included were spontaneous ADR reports with a fatal outcome.

### 2.2. Variables

The identified DRDs were characterized by demographic and clinical variables, and suspected drugs. The involved drugs of the DRD were classified using the Anatomical Therapeutic Chemical (ATC) classification system [19]. The characteristics of the ADRs, the number of involved drugs, indication, dosage and treatment duration, were extracted. We defined polymedication as when a patient received at least 10 drugs at the moment of the ADR. We classified the period between the drug start date and the reaction start date as the following: ‘acute’ when the drug was started within the week before, ‘subacute’ when it was started between 1 week and 6 months before, and ‘chronic’ when it was started more than 6 months before.

If a drug–drug interaction was suspected a review of the literature was conducted to document the interaction. We classified them as either pharmacodynamic or pharmacokinetic. Pharmacodynamic interactions were defined as those in which drugs influence each other´s pharmacologic effect, even more it was evaluated if they were synergistic or antagonistic. Pharmacokinetic interactions were defined as those in which a drug may result in the increase or the decrease in plasma drug concentrations [20].

Drugs were classified as being ‘under additional monitoring’ or ‘not’ to identify recently marketed drugs or drugs with safety concerns. ‘Additional monitoring’ is a term denoted by the European Medicines Agency to medicines that are more intensively monitored than others. This is generally because there is less safety information available, for example because the medicine has been recently approved or there is limited data on its long-term use. The ‘additional monitoring’ aims to enhance reporting of suspected ADRs for medicines in which the clinical evidence base is less well developed to collect information as early as possible to further inform the safety of these medicines when used in everyday medical practice [21]. The involved drugs in our study were denoted as ‘under additional monitoring’ if they were included in the EMA’s list of medicines under additional monitoring according to the year in which the ADR occurred [22].

ADRs were classified as type A or type B reactions. Type A reactions depend on the mechanism of action of the drug, are dose-dependent, predictable and preventable; whereas type B reactions do not depend on the drugs mechanism of action, are not dose-dependent, could be idiosyncratic or hypersensitivity-related, and unpreventable [23] Two evaluators (A.L.A. and E.M.) assessed type of reactions for each DRD case and a consensus was reached when discrepancies between scores were present.

### 2.3. ADR Causality Assessment

To assess the likelihood of whether a suspected ADR is due to a drug the Naranjo algorithm was used [24]. According to the algorithm the ADRs were classified as definite (9–12 points), probable (5–8 points), possible (1–4 points) or doubtful (0 points). Also, the involved drugs in the DRD were classified as either cause of or contributor to death. The term cause of death was used when the drug was directly related to the death and contributor when a cofactor concomitantly precipitated the death.

Two evaluators (A.L.A. and E.M.) assessed both the causality method and contribution of death for each DRD case. Consensus was reached when discrepancies were present.

### 2.4. Preventability of ADR

The Schumock and Thornton criteria were applied to identify preventable DRD [25]. An ADR was considered ‘preventable’ when answering “yes” to one or more of the following questions: (i) was the drug inappropriate for the patient’s condition? (ii) were the dose, frequency and route of administration inappropriate for the patient’s age, weight or disease state? (iii) therapeutic drug monitoring or other necessary laboratory test were not performed; (iv) was there a history of allergy or previous reaction to the drug? (v) was a drug interaction involved? (vi) was a toxic serum drug level documented? and (vii) was poor compliance involved in the ADR?

Two evaluators (A.L.A. and E.M.) assessed the preventability for each DRD case and a consensus was reached when discrepancies were present.

### 2.5. Comparison of Two Pharmacovigilance Studies Assessing DRD Cases

As a part of active surveillance method and for purposes of a comparison of different pharmacovigilance methods, we included the results of a previously published retrospective study from our investigation group [26]. In this study, we selected potential DRD cases from the hospital´s mortality registry of the year 2015 if the patient´s death diagnosis potentially oriented to a death caused by a drug (such as acute renal failure, cardiac arrhythmia, hemorrhage, infections, pancreatitis, or hepatitis). The potential DRD cases were selected for further review. All potential cases were assessed to determine whether the death was ‘possibly’ or ‘probably’ related to a drug. Any drug-related case was included in the study and considered as an actual DRD.

We compared the results of the characteristics of spontaneous reporting DRD cases vs. DRD cases of a previous retrospective study. The results and comparisons of the variables of both studies have been described in the corresponding tables.

### 2.6. Statistical Analysis

Descriptive analyses were used to assess the DRD cases. Summary statistics are presented as percentages for categorical variables and as medians (range) for continuous variables.

To compare the characteristics of DRD between cases from spontaneous reporting and from the retrospective study, Pearson chi-square or Fisher exact test was used for categorical variables and Mann–Whitney for numerical continuous variables. All calculations were performed using STATA version 14 (StataCorp. 2015. Stata Statistical Software: Release 14. College Station, TX, USA: StataCorp LP.)

## 3. Results

The characteristics of the DRD spontaneously reported are described in detail.

From a total of 1,460 ADRs registered to the hospital pharmacovigilance program in a 10-year period, 73 (5%) were DRDs, identified by the SR method. The median age was 75 years (range: 1 month–94), of whom 60.3% (44/73) were men. The median number of drugs during the ADR episode per patient was nine (range 2–17), and the median hospital stay was 8 days (range 1–90) (see Table 1).

### 3.1. Characteristics of Fatal ADR

ADR was the cause of hospital admission in 50 cases (68.5%, 50/73), and the ADR started during hospitalization in 23 patients (31.5%, 23/73) (see Table 1).

The most frequent DRDs were hemorrhage (30 cases, 41.1%), followed by infections (13 cases, 17.8%). The remaining types of DRD are presented in Table 2.

The most frequent DRD infections were sepsis (46.2%, 6/13) in drug-immunosuppressed patients, and lung infections (30.8%, 4/13). Detailed information for all the DRDs is presented in Table 3.

### 3.2. Characteristics of Drug Related to ADR

In 42 (57.3%, 42/73) DRD cases, there was only one drug involved, two suspected drugs in 19 (26.0%, 19/73), three drugs in seven (9.6%, 7/73), and four or more different related drugs in five cases (6.8%, 1/73). In 29 (39.7%, 29/73) DRD cases, a drug–drug interaction was present; all were classified as pharmacodynamic and synergistic interactions, except one, which was a pharmacokinetic interaction between amiodarone and acenocoumarol. The suggested mechanism could be that amiodarone is a potent inhibitor of CYP2C9, CYP1A2, and CYP3A4, which are largely responsible for acenocoumarol metabolism, thus amiodarone will increase its plasma concentrations, thereby increasing the patient´s INR and augmenting the risk of cerebral hemorrhage [27]. The number of DRD patients with polymedication was 30 (41.0%, 30/73).

In total, there were 123 involved drugs for the 73 DRD cases. Thirty-seven drugs (30.0%, 37/123) were classified in ATC category B, 33 (26.3%, 33/123) in category L, 12 (9.8%, 12/123) in category J, and 11 (8.9%, 11/123) in category H. The remaining involved drugs by category are detailed in Table 4.

The most involved drugs in the B category were acenocoumarol (16/37) and acetylsalicylic acid (12/37). From the category L, mycophenolic acid (4/33) and tacrolimus (3/33) were more frequently involved. Amoxicillin/clavulanic acid (3/12) and ceftriaxone (2/12) were the most frequent involved drugs in the anti-infective category. Prednisone (7/11) was the most frequent drug involved in the H category. The drugs involved in DRD are detailed in Table 5. The association between the type of DRD and drug involved is detailed in Table 6.

The involved drugs were started within the week before the ADR, referred to as subacute, in 23 patients (31.5%, 23/73). Subacute treatment was identified in 21 patients (28.8%, 21/73) and chronic in 28 (38.4%, 28/73). In one DRD case, the beginning of the drug was unknown (1.4%, 1/73) (see Table 1).

Drugs were the cause of death in 42 DRD cases (57.5%, 42/73) and contributed to death in 31 patients (42.4%, 31/73). The main contributive causes for the DRD with cerebral hemorrhages were due to blunt head trauma (16.4%, 12/73) or secondary to a hypertensive episode (12.3%, 9/73) (see Table 1).

The involved drugs were under additional monitoring in 8.90% of the DRD cases (8/73). These drugs were atezolizumab, canagliflozin, durvalumab, temozolomide, gefitinib, obinutuzumab, osimertinib, and rivaroxaban.

Further, 28.7% (21/73) of the DRDs were classified as being type B, and 7.2% (52/73) were type A reactions (see Table 1).

### 3.3. ADR Causality Assessment

The median Naranjo score of DRD cases was 4 (range 3–7). ADRs were classified into the following two categories: possible (score range 1–4; 41/73 ADR, 56.2%) and probable (score range 5–8; 32/73 ADR, 43.8%) (see Table 7).

### 3.4. Preventability of ADR

According to the Schumock and Thornton criteria, DRDs were potentially preventable in 36 cases (49.3%, 36/73) (see Table 7). The most frequent criteria were the presence of a drug–drug interaction in the DRD (29 cases, 82.8%) (see Table 8).

The drug–drug interactions detected by the criteria as potentially preventable were mostly pharmacodynamic with a synergistic effect. Thirty-one percent (9/29) were hemorrhages, and the involved drugs were a combination of anticoagulants and antithrombotics, or double antiplatelet therapy. Further, 41.4% (12/29) were infections, mostly in immunocompromised patients receiving combination chemotherapy for cancer with corticoids.

The second most selected criteria was ‘the drug was not appropriate for the patient’s condition’ in four DRD cases (see Table 8); in two cases there was no clinical indication in the patient’s chart to receive the drug (both cases acetylsalicylic acid). The other cases were ceftriaxone in a patient with an active episode of severe cholestasis that, after receiving the drug, worsened, and metformin in a hospitalized diabetic and renal impaired patient, which was not withdrawn from the prescription, even though the patient had renal failure that associated with a metabolic acidosis. Also, in the criteria, ‘the dose, frequency, and route of administration were inappropiate for the patient´s age, weight or disease state’, in one case the dose of cefepime was not adjusted to renal function in a renal-impaired patient that induced encephalopathy.

### 3.5. Comparison of DRD from Spontaneous Reporting vs. Retrospective Study on Diagnosis at Death

When both studies were compared, there were several differences. The spontaneous reporting study reported more type B reactions (28.8% vs. 0%; *p* < 0.0001), more hospital-acquired DRD (31.5% vs. 8.2%; *p* < 0.001), higher number of concomitant drugs (9 vs. 7; *p* < 0.0035), and more drugs started within the week before the ADR (31.5% vs. 11%; *p* = 0.002) than the retrospective study (see Table 9).

When the types of DRD were compared, hemorrhagic and infection reactions were the most frequent in both the studies. However, the spontaneous reporting study identified hepatobiliary, pulmonary, and allergic reactions, while the retrospective study did not (*p* = 0.043, *p* = 0.029, *p* = 0.012, respectively). On the other hand, the retrospective study did identify more infection reactions than spontaneous reporting (*p* = 0.001) (see Table 10).

When the suspected drugs were compared, the spontaneous reporting study identified drugs involved in the DRD cases in the J category (anti-infectives), and in the A category (mainly antidiabetics and proton pump inhibitors) were those that the retrospective study did not identify (*p* = 0.0030 and *p* = 0.0006, respectively) (see Table 11).

When comparing the causality assessment and preventability criteria of the DRD, no significant differences were found (see Table 12).

## 4. Discussion

As DRDs are an important cause of mortality [11,28,29], the present study was performed to describe the clinical and drug characteristics of DRD spontaneously reported from a hospital pharmacovigilance program. As is known in spontaneous reporting methods, under-reporting is a common problem [30]. Additional pharmacovigilance methods, such as active surveillance studies, may increase the rate of ADR detection [17]. The results of our study show an increase in number. Overall, the one-year retrospective study detected additional DRDs to the spontaneous reporting. Although the number of DRDs in both methods was the same, the study time was different (one year vs. 10 years).

The spontaneous reporting study notified more hospital-acquired DRD, probably because of the nature of an in-hospital pharmacovigilance program, and because the health professional that notified the ADR is directly involved in the wellbeing of the patient. When comparing the time from the start of the involved drugs, the spontaneous reporting study identified more drugs that were started within the week before the ADR that could correlate with patients having the DRD during hospital admission.

It is interesting to point out that from both pharmacovigilance studies, the most common DRDs were hemorrhages and infections, which supports the findings in a recent retrospective analysis of VigiBase, the largest pharmacovigilance database in the world, from the World health Organization, [31] and a meta-analysis of observational studies in European hospitals [12]. Therefore, the most common involved drugs were anticoagulants and/or anti-thrombotics, and antineoplastic agents [12,31]. Preventive measures should be implemented to promote and improve closer follow-up of anticoagulant use.

The spontaneous reporting study also detected that anti-infectives for systemic use were frequently involved drugs in DRD, similar to the results obtained from a retrospective study from the Italian pharmacovigilance database, where systemic anti-infective drugs, and antineoplastic and immunomodulating agents were the most involved drugs of ADR with fatal outcome [32]. In both methods, drug–drug interactions were linked to almost half of DRD cases, very similar to a meta-analysis [12] and a prospective study [28]. Most of the interactions detected were pharmacodynamic and synergistic interactions.

In general, from pharmacovigilance systems, it is urged that health professionals notify rare, unknown and/or serious ADRs to spontaneous reporting programs [30]. This may explain why the spontaneous reporting study identified rare and bizarre type B reactions and the retrospective study did not. Moreover, type A ADRs are probably more detectable to the trained eye of a clinical pharmacologist and specialized nurses.

In both pharmacovigilance studies, only 6–10% of the drugs were classified as being on additional safety monitoring, and the results of this study indicate that older drugs continue to be most commonly implicated in DRD, similar to another study [28].

The results of both studies indicate that DRDs are more commonly observed in male patients over 65 years old, which supports the findings of the retrospective analysis of Vigibase [31].

We used the Naranjo algorithm for causality assessment; although it is one of the most frequently used worldwide, limitations are present not just when assessing dead patients, but in general, such as drug rechallenge, because it often does not occur in the “real world” of clinical practice. This might not occur in serious ADRs, since rechallenge might be considered unethical and may pose a considerable risk to the patient [33].

Almost half of the DRDs were preventable, in both strategies, similar to previous studies [17]. The most frequent Schumock and Thornton criterion met was that the ADR involved a documented drug–drug interaction. As stated before, most of the interactions detected were pharmacodynamic and synergistic interactions from known interactions, and are therefore likely to be preventable. However, in diseases such as cancer, combined therapies are necessary to produce effectiveness of drugs, due to the synergic interaction [34]. In these cases, deaths were probably unavoidable. However, there were a few cases that, in theory, could have been prevented. For example, a follow-up by health professionals of the drugs taken by patients, to see if they still need it or if it is really indicated, could reduce the risk of DRD. An effective measure could be in-hospital computed-based prescription to alert prescribers of contraindicated drugs if the patient has renal failure, or block a prescription if the recommended duration has expired.

### Strengths and Limitations

The main limitation of the present study is the design, being a single-center and retrospective analysis. The differences between reporting ADR strategies implemented in different hospitals and countries, and the variations in characteristics of patients and individual susceptibilities for reporting is why the results from the present study reflect DRD for a tertiary hospital and not in the general population. The design of a prospective study, tailored to collect specific data, may be more complete than a retrospective study, but one disadvantage is the long follow-up period required for events or diseases to occur. Instead, retrospective cohort studies are better indicated, given the timeliness and inexpensive nature of the study design [35]. Considering that the event in this study was death, it will greatly complicate a prospective follow-up.

Another limitation resides on the ADR itself, notified by health professionals, which may reflect certain characteristics from a specific hospital pharmacovigilance program influenced by the specialist or interests of health professionals that form part of it. Therefore, it is difficult to extrapolate the results and compare them with other pharmacovigilance programs, but the results are similar in other studies [12].

One of the strengths of this study was that two evaluators assessed some of the study variables, such as the type of the ADR, the causality assessment, and preventability of DRD cases, and a consensus was reached when discrepancies between the scores were present. Moreover, most available studies about SR are from national pharmacovigilance centers, which usually lack specific data related to ADR. To our knowledge, this study is the first European study assessing and describing DRD from a hospital pharmacovigilance program, and comparing two different methods of pharmacovigilance to assess DRD.

## 5. Conclusions

When assessing the characteristics of DRD cases identified by spontaneous reporting in our hospital, we conclude that the results of this study indicate that about 10% of the drugs in DRD had limited safety data (were under additional monitoring), a third of the DRD included a drug–drug interaction (synergistic) from known interactions, and almost half of the DRDs were potentially preventable according the Schumock and Thornton criteria. Preventable DRD should be accurately assessed in further studies, and preventive measures should be implemented in clinical practice.

By comparing different pharmacovigilance studies, the results of this study suggest that the different methods can be complementary to increase the level of information, and thus be able to increase the safety of drugs for in-hospital patients. It is important to note that to achieve this objective, hospital pharmacovigilance programs should have appropriate resources to carry out this function. We conclude and reinforce that the combination of pharmacovigilance methods is mandatory to detect, assess, understand, and design strategies to prevent ADR in a hospital setting, to ensure patient safety.

## Figures and Tables

**Table 1 jcm-10-04053-t001:** Characteristics of patients with DRD.

**Characteristics of Patients with DRD**	
Age: years (median, range)	75 (1 month–94)
Sex: male, *n* (%)	44 (60.3%)
Polymedication, *n* (%)	29 (39.7%)
Number of concomitant drugs (median, range)	9 (2–17)
Hospital stay: days (median, range)	8 (1–90)
**Characteristics of ADR**	
Hospital Admission, *n* (%)	50 (68.5%)
Hospital-acquired DRD, *n* (%)	23 (31.5%)
Drug–drug Interactions	29 (39.7%)
Drugs under additional monitoring	8 (10.9%)
Number of suspected drugs (median, range)	1 (1–5)
**Type of reaction**	
Type A reactions, *n* (%)	52 (71.2%)
Type B reactions, *n* (%)	21 (28.8%)
**Cause or contribution to death**	
Drugs were the cause of death, *n* (%)	42 (57.5%)
Drugs contributed to death, *n* (%)	31 (42.5%)
Time from the start of the involved drugs ^1^	
Subacute (≤1 week)	23 (31.5%)
Acute (>1 week–6 months)	21 (28.7%)
Chronic (>6 months)	28 (38.3%)

^1^ Data not available in one patient.

**Table 2 jcm-10-04053-t002:** Types of DRD.

Type of DRD	*N* (%)
Hemorrhagic alterations	30 (41.1%)
Infections	13 (17.8%)
Pulmonary alterations	7 (9.6%)
Allergic reactions	6 (8.2%)
Cardiovascular alterations	5 (6.8%)
Hepatobiliary alterations	4 (5.4%)
Hematological alterations	2 (2.7%)
Renal alterations	1 (1.4%)
Neurological alterations	1 (1.4%)
Endocrinological alterations	1 (1.4%)
Skin alterations	1 (1.4%)
Others	2 (2.7%)
Total	73 (100%)

**Table 3 jcm-10-04053-t003:** Type of DRD.

Type of DRD	*N* (%)
Hemorrhagic alterations	30 (41.1%)
Cerebral	23
Pulmonary	3
Gastrointestinal	2
Renal, Peritoneal (1 each)	2
Infections	13 (17.8%)
Sepsis	6
Lung infections	4
Pseudomembranous colitis	2
Progressive multifocal leukoencephalopathy	1
Pulmonary alterations	7 (9.6%)
Interstitial pneumonitis	4
Pulmonary Fibrosis, Bilateral pulmonary embolism, Pulmonary bronchospasm (1 each)	3
Allergic reactions	6 (8.2%)
Anaphylactic shock	3
Anaphylaxis	2
Acute epiglottitis	1
Cardiovascular alterations	5 (6.8%)
QT interval elongation, Ventricular fibrillation, Dilated cardiomyopathy, Myocarditis, Cardiac arrest (1 each)	5
Hepatobiliary alterations	4 (5.54%)
Cholestasis and pancreatitis, Hepatic venous occlusion syndrome, Acute Hepatitis, Liver failure (1 each)	4
Hematological alterations	2 (2.7%)
Aplastic anemia, Bicytopenia (1 each)	2
Others	6 (8.2%)
Acute renal failure, DRESS syndrome, Encephalopathy with epileptic status, Hypoglycemic coma, Lactic acidosis, Multiple drug intoxication (1 each)	6
Total	73 (100%)

**Table 4 jcm-10-04053-t004:** The Anatomical Therapeutic Chemical (ATC) classification system of involved drugs.

ATC Category	Therapeutic Area	*N* (%)
B	Blood and blood-forming organs	37 (30.0%)
L	Antineoplastic and immunomodulating agents	33 (26.3%)
J	Anti-infectives for systemic use	12 (9.8%)
H	Systemic hormonal preparations, excluding sex-hormones and insulins	11 (8.9%)
A	Alimentary tract and metabolism	9 (7.3%)
N	Nervous system	9 (7.3%)
C	Cardiovascular system	5 (4.1%)
M	Musculoskeletal system	3 (2.4%)
G	Genito-urinary system and sex hormones	1 (0.8%)
P	Antiparasitic products, insecticides and repellents	1 (0.8%)
R	Respiratory system	1 (0.8%)
V	Various	1 (0.8%)
	Total	123

**Table 5 jcm-10-04053-t005:** Drugs involved in DRD.

Involved Drug	ATC Category	*n* (%)
Acenocoumarol	B	16 (13.0%)
Acetylsalicylic acid	B	12 (9.8%)
Prednisone	H	7 (5.7%)
Mycophenolate mofetil	L	4 (3.3%)
Amoxicillin/clavulanic acid	J	3 (2.4%)
Clopidogrel	B	3 (2.4%)
Enoxaparine	B	3 (2.4%)
Tracolimus	L	3 (2.4%)
Amiodarone	C	2 (1.6%)
Cefrtiaxone	J	2 (1.6%)
Citarabine	L	2 (1.6%)
Dexamethasone	H	2 (1.6%)
Docetaxel	L	2 (1.6%)
Metformin	A	2 (1.6%)
Metamizole	N	2 (1.6%)
Olanzapine	N	2 (1.6%)
Omeprazole	A	2 (1.6%)
Acetaminophen, Amikacin, Allopurinol, Alprazolam, Apixaban, Atezolizumab, Azatioprine, Beclomethasone, Bupivacaine, Bleomycin, Canagliflozin, Cefuroxime, Ceftazidime, Cefepime, Cetuximab, Cotrimoxazol, Ciclosporine, Contrast media, Dacarbazine, Dexketoprofen, Doxorubicin, Durvalumab, Edoxaban, Epinephrine, Etanercept, Etoposid, Gefitinib, Glibenclamide, Glicazide, Hydrochlorothiazid, Hydrxychloroquine, Hydroxyurea, Ibuprofen, Idarubicin, Infliximab, Leflunamide, Losartan, Mesna, Metilprednisolone Metotrexate, Moxifloxacin, Obinutuzumab, Ondasentron, Osimertinib, Pemetrexed, Rivaroxaban, Sertraline, Sitagliptine, Temozolomide, Tremelimumab, Vancomycin, Vinblastine, Zolpidem		1 each; 54 (0.8% each)
Total		123 (100%)

**Table 6 jcm-10-04053-t006:** Association between type of DRD and drug involved.

Type of DRD (*n*)	Drugs Involved (ATC)
Hemorrhagic alterations (30)	
Cerebral (23)	Acetylsalicylic acid (B), Acenocoumarol (B), Amiodarone (C), Apixaban (B), Clopidogrel (B), Enoxaparin (B), Rivaroxaban (B)
Pulmonary (3)	Acetylsalicylic acid (B), Clopidogrel (B), Enoxaparin (B), Ibuprofen (M)
Gastrointestinal (2)	Acenocoumarol (B), Dexketoprofen (M), Enoxaparine (B)
Renal, Peritoneal (1 each)	Acenocoumarol (B), Acetylsalicylic acid (B)
Infections (13)	
Sepsis (6)	Azatioprine (L), Cetuximab (L), Ciclosporin (L), Cytarabine (L), Dexamethasone (H), Docetaxel (L), Etanercept (L), Idarubicin (L), Infliximab (L), Methylprednisolone (H), Mycophenolate acid (L), Prednisone (H)
Lung infections (4)	Leflunomide (L Methotrexate (L), Mycophenolate acid (L), Prednisone (H), Tacrolimus (L)
Pseudomembranous colitis (2)	Amikacin (J), Amoxicillin/clavulanic acid (J), Cefuroxime (J), Ceftazidime (J), Omeprazole (A)
Progressive multifocal leukoencephalopathy (1)	Obinutuzumab (L)
Pulmonary alterations (7)	
Pulmonary Fibrosis (1)	Docetaxel (L)
Interstitial pneumonitis (4)	Amiodarone (C), Bleomycin (L), Dacarbazine(L), Doxorubicin(L), Gefitinib (L), Osimertinib(L), Prednisone (H), Vinblastine(L),
Bilateral pulmonary embolism (1)	Oral anticonceptives (G)
Pulmonary bronchospasm (1)	Mesna (R)
Allergic reactions (6)	
Anaphylactic shock (3)	Ceftriaxone (J), Metamizol (N), Moxifloxacin (J), Vancomycin (J)
Anaphylaxis (2)	Amoxicillin/clavulanic acid (J), Metamizole (N)
Acute epiglottitis (1)	Amoxicillin/clavulanic acid (J)
Cardiovascular alterations (5)	
QT interval elongation (1)	Olanzapine (N), Sertraline (N)
Ventricular fibrillation (1)	Bupivacain (N), Epinephrine (C)
Dilated cardiomyopathy (1)	Hydroxychloroquine (P)
Myocarditis (1)	Durvalumab (L), Tremelimumab (L)
Cardiac arrest (1)	Ondasentron (A)
Hepatobiliary alterations (4)	
Cholestasis and pancreatitis (1)	Ceftriaxone (J)
Hepatic venous occlusion syndrome (1)	Citarabine (L), Etoposid (L), Hydroxyurea (L)
Acute Hepatitis (1)	Atezolizumab (L)
Liver failure (1)	Canagliflozin (A), Edoxaban (B), Gliclazide (A), Sitagliptine (A)
Hematological alterations (2)	
Aplastic anemia (1)	Cotrimoxazole (J), Temozolomide (L)
Bycitopenia (1)	Pemetrexed (L)
Others (6)	
Acute renal failure, DRESS syndrome, Encephalopathy with epileptic status, Hypoglycemic coma, Lactic acidosis, Multiple drug intoxication (1 each)	Acetaminophen (N), Allopurinol (M), Alprazolam (N), Cefepime (J), Contrast media (V), Glibenclamide (A), Hydrochlorothiazide (C), Losartan (C), Metformine (A), Olanzapine (N), Zolpidem (N)

**Table 7 jcm-10-04053-t007:** DRD causality assessment and preventability.

Naranjo Algorithm	*N* (%)
Probable	32 (43.8%)
Possible	41 (56.2%)
Schumock and Thornton criteria	
Potentially preventable	36 (49.3%)
Potentially not preventable	37 (50.7%)

**Table 8 jcm-10-04053-t008:** Frequency of DRD cases meeting the Schumock and Thornton criteria.

Schumock and Thornton Criteria	Number of DRD Cases Meeting the Criteria
(1) the drug was not appropriate for the patient’s condition	4
(2) the dose, frequency, and route of administration were inappropriate for the patient’s age, weight or disease state	1
(3) therapeutic drug monitoring or other necessary laboratory test was not performed	0
(4) the patient had a history of allergy or previous reaction to the administered drug	1
(5) a documented drug interaction was involved in the ADR	29
(6) a serum concentration above the therapeutic range was documented	1
(7) noncompliance was involved in the ADR	0

**Table 9 jcm-10-04053-t009:** Comparison of patients and DRD characteristics.

	Spontaneous Reporting	Retrospective Study	*p* Value
Characteristics of patients with DRD			
Age: years (median, range)	75 (1 month–94)	72 (19–94)	0.535
Sex: male, *n* (%)	44 (60.3%)	53 (72.6%)	0.115
Polymedication, *n* (%)	29 (39.7%)	32 (43.8%)	0.615
Number of concomitant drugs (median, range)	9 (2–17)	7 (2–14)	0.003
Hospital stay: days (median, range)	8 (1–90)	5 (0–57)	0.131
Characteristics of ADR			
Hospital Admission, *n* (%)	50 (68.5%)	67 (91.8%)	<0.001
Hospital-acquired DRD, *n* (%)	23 (31.5%)	6 (8.2%)	
Drug–drug Interactions	29 (39.7%)	32 (43.8%)	0.615
Drugs under additional monitoring	8 (10.9%)	5 (6.8%)	0.383
Number of suspected drugs (median, range)	1 (1–5)	1 (1–4)	0.882
Type of reaction			
Type A reactions, *n* (%)	52 (71.2%)	73 (100%)	<0.0001
Type B reactions, *n* (%)	21 (28.8%)	0	
Cause or contribution to death			
Drugs were the cause of death, *n* (%)	42 (57.5%)	38 (52.1%)	0.506
Drugs contributed to death, *n* (%)	31 (42.5%)	35 (74.9%)	
Time from the start of the involved drugs ^1^			
Subacute (≤1 week)	23 (31.5%)	8 (11%)	0.002
Acute (>1 week–6 months)	21 (28.7%)	27 (38%)	0.290
Chronic (>6 months)	28 (38.3%)	36 (51%)	0.182

^1^ Data not available in one patient in the spontaneous reporting and in two in the retrospective study.

**Table 10 jcm-10-04053-t010:** Comparison of type of DRD.

Type of DRD	Spontaneous Reporting	Retrospective Study	*p* Value
Hemorrhagic alterations	30 (41.1%)	34 (46.5%)	0.505
Infections	13 (17.8%)	32 (43.8%)	0.001
Pulmonary alterations	7 (9.6%)	1 (1.4%)	0.029
Allergic reactions	6 (8.2%)	0	0.012
Cardiovascular alterations	5 (6.8%)	2 (2.7%)	0.245
Hepatobiliary alterations	4 (5.4%)	0	0.043
Hematological alterations	2 (2.7%)	0	0.154
Renal alterations	1 (1.4%)	1 (1.4%)	0
Neurological alterations	1 (1.4%)	1 (1.4%)	0
Endocrinological alterations	1(1.4%)	0	1.007
Skin alterations	1 (1.4%)	0	1.007
Others	2 (2.7%)	2 (2.7%)	0
Total	73 (100%)	73 (100%)	

**Table 11 jcm-10-04053-t011:** Comparison of the ATC classification system of involved drugs.

ATC Category	Therapeutic Area	Spontaneous Reporting	Retrospective Study	*p* Value
B	Blood and blood-forming organs	37 (30.0%)	38 (32.8%)	0.6558
L	Antineoplastic and immunomodulating agents	33 (26.3%)	46 (39.6%)	0.0351
J	Anti-infectives for systemic use	12 (9.8%)	0	0.0006
H	Systemic hormonal preparations, excluding sex-hormones and insulins	11 (8.9%)	21 (18.1%)	0.0377
A	Alimentary tract and metabolism	9 (7.3%)	0	0.0030
N	Nervous system	9 (7.3%)	5 (4.3%)	0.3225
C	Cardiovascular system	5 (4.1%)	5 (4.3%)	
M	Musculoskeletal system	3 (2.4%)	1 (0.9%)	0.3422
G	Genito-urinary system and sex hormones	1 (0.8%)	0	0.9668
P	Antiparasitic products, insecticides and repellents	1 (0.8%)	0	0.9668
R	Respiratory system	1 (0.8%)	0	0.9668
V	Various	1 (0.8%)	0	0.9668
	Total	123	116	

**Table 12 jcm-10-04053-t012:** Comparison of causality assessment and preventability.

	Spontaneous Reporting	Retrospective Study	*p* Value
Naranjo Algorithm			
Probable	32	27	0.339
Possible	41	46	
Schumock and Thornton criteria			
Potentially preventable	36	34	0.740
Potentially not preventable	37	39	

## Data Availability

The data presented in this study are available on request from the corresponding author.

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
