# Peer review of "Drug-Related Deaths in a Tertiary Hospital: Characteristics of Spontaneously Reported Cases and Comparison to Cases Detected from a Retrospective Study"

_jcm, 2021, doi:10.3390/jcm10184053_

Round 1
Reviewer 1 Report
Thank you for giving me the possibility of reweving the present paper.
The manuscript focuses on cases of drug-related deaths detected in hospitalised patients during the course of 10years. Interestingly, two pharmacovigilance methods have been used to compare data.
The manuscript is interesting and in general well-written, but please consider the following comments for improving it.
Introduction: please improve this section, focusing on DRDs and what is the background of the study. Moreover, as data should be compared with previous ones, please specify in the Introduction how this study would improve the previous one, and what could you expect. The Introduction, Better the Methods section, should explain the reader the two pharmacovigilance methods used.
Discussion: regarding the limitations of the study, one consideration focuses on the data collected. I wonder if you have tried to compare your data to the European ones. On this regard, EudraVigilance is the system for managing and analysing information on suspected adverse reactions (AR) to medicines authorised in the European Economic Area (EEA). It would be interesting to understand if your results are consistente with them.
Reviewer 2 Report
Dear Authors,
I have read with great interest this manuscript, as the topic is of high interest, especially if we think about the preventable drug-related deaths (DRD). I have some doubts, comments and suggestions:
- The title: perhaps it could be more specific. Are the authors describing drug-related deaths in the hospital? Are they additionally comparing two pharmacovigilance methods? Or are they comparing the characteristics of DRD reports obtained by two pharmacovigilance methods? Perhaps the title should specify it.
- Abstract: the first sentence is difficult to understand.
- The definition of ADR has evolved on time. Although this is the first one (1972), perhaps more recent definitions should be included (please, cf. Edwards IR, Aronson JK. Adverse drug reactions: definitions, diagnosis, and management. Lancet. 2000 Oct 7;356(9237):1255-9. doi: 10.1016/S0140-6736(00)02799-9. PMID: 11072960.)
- Although the active PV study had already been published, the reader could benefit of a more detailed explanation in the Methods section (for example, the type of cause of death included, etc.)
- I would suggest the authors to split the Methods section in two subsections, to detail the procedures of both studies.
- Regarding the spontaneous reporting study, it should be described the selection criteria of the reports (to obtain 73 reports out of 1,460 reports). Were all reported deaths included? Only those with an established causal relationship with the medicine? Additionally, the selection criteria for the SR study, were comparable to the inclusion criteria used in the AS study? All this should be described in the Methods section.
- “Polymedication was defined as when a patient received at least 10 drugs at the moment of the ADR”. Is this criterion well defined? (Please add the appropriate reference). It seems that 10 medicines to define polymedication is quite high.
- Please, cite the reference for the classification “acute”, “subacute” and “chronic”, for the time of use in the case of medicines.
- The concept ‘drugs being under additional monitoring’ is probably an internal jargon, the readers cannot understand it. Please, define this concept and specify which medicines included during the study period.
- The 1977 Rawlins and Thompson classification (A and B) is quite useful, but incomplete, as it has been discussed over time). To be noted, it is difficult to classify some ADRs by using this binary classification. So, the authors should describe how many reports could not be classified because of the limitations of the method (please, cf. Aronson JK, Ferner RE. Joining the DoTS: new approach to classifying adverse drug reactions. BMJ. 2003;327(7425):1222-1225. doi:10.1136/bmj.327.7425.1222 or other more recent articles on this topic.
- In subsection 2.3, the manuscript says: “The Naranjo Algorithm was used to assess the causal probability [22]. This algorithm determines the likelihood of whether an ADR is due to the drug.”. In my opinion, an algorithm cannot determine anything. It is just a tool to help in the assessment of the likelihood that a suspected ADR is due to a medicine.
- The results start by describing the reports selected from the SR study, but Table 1 already includes data from the AS study. Perhaps a paragraph should describe this second study, before starting presenting numbers without understanding its origin.
- Why does Table 3 not include information from the AS study?
- In different parts of the study, the Authors refer to drug-drug interactions. My question is, how were these drug-drug interactions defined? Meaning, were “potential” (i.e., theoretical) drug-drug interactions? Or proven drug-drug interactions? This should also be clarified in the Methods section.
- In Table 4: Although the alphabetical order is appealing, in my opinion, to order the therapeutic areas according to the number of reports identified, would help to focus on the biggest numbers (groups B and L), as many of the other groups are marginal. It could also help to focus on groups J and A: why no report from the AS study included medicines belonging to J or A? (e.g., perhaps the inclusion criteria for the AS study made less frequent the use of any of these medicines by patients?)
- Why does Table 5 not include the column for the AS study?
- Is Table 6 referring to the SR study? the AS study? Both?
- In my opinion, one of the most relevant results of the study is the preventability of the detected DRB. Approximately 50% of the reports in both studies could have been avoided (in theory). This is, theoretically, 50% of the patients who died, could have been saved.
So, my suggestion is that the characteristics of the patients (drugs involved, reactions, time of use, etc) are compared for both studies, according to the “potentially preventable” or the “potentially not preventable” label. From this, perhaps the team could be able to identify which are the characteristics of the “preventable” ADRs, and this knowledge helps to implement measures to change the current situation. For example, if haemorrhages x anticoagulants are “preventable”, what could the healthcare personnel do to reduce mortality due to this cause?
- The initial sentence of the Discussion says: “The present study was performed to describe the clinical and drug characteristics of DRD in a hospital setting.”, but the paragraph describing the aim of the study says: “The main objective of this study was to describe the characteristics of DRD from a Hospital Pharmacovigilance Program that include ADR spontaneously reported by health professionals and to compare them to those of an active surveillance method carried out by clinical pharmacologists from a systematic selection of DRD cases.”
I would suggest making clear which is the aim.
In my opinion, the paragraph “It is interesting to point that from both pharmacovigilance methods the most common DRD were heamorrhages and infections, which support the findings in a retrospective analysis of VigiBase, the World health Organization pharmacovigilance database,[28] and a meta-analysis of observational studies in European hospitals [13]. Therefore, the most common involved drugs were anticoagulants and/or antithrombotics and antineoplastic agents [13,28]” is the key finding of this study. I would suggest the authors to work from it, as this could be one of the most interesting leading messages from their experience.
As I see, it, the point is that, in a third-level hospital, different PV methods have been conducted. An analysis of the DRD obtained from both methods has found that the results are not so different in many variables and have found that the most common DRD were haemorrhages and infections involving anticoagulants, antithrombotic and antineoplastic agents. But, above all, almost half of the DRD detected, were preventable.
This pattern has been seen in other studies and places. So, the remaining question is why these well-known ADR continue to cause deaths in hospitals, and what could we do to reduce them?
Round 2
Reviewer 2 Report
Dear Authors,
Thank you for your work to address the comments and suggestions made by the Reviewers. In my opinion, the new version of the manuscript makes clear what has been done, and the results of the research and its discussion are suitable with the described methods. The new tables are also clear.